# Health Disparities and Inequities in the Utilization of Proton Therapy for Prostate Cancer

**DOI:** 10.3390/cancers16223837

**Published:** 2024-11-15

**Authors:** Cyrus Gavin Washington, Curtiland Deville

**Affiliations:** 1Department of Radiation Oncology, University of Miami-Sylvester Comprehensive Cancer Center, Jackson Memorial Hospital, Miami, FL 33136, USA; cyrus.washington@med.miami.edu; 2Department of Radiation Oncology and Molecular Radiation Sciences, Johns Hopkins University School of Medicine, Baltimore, MD 21205, USA

**Keywords:** prostate cancer, proton, disparities, IMRT, race, insurance, socioeconomic, inequities

## Abstract

As technology progresses, the scope and practice of medicine adapts, resulting in treatment applications that improve the precise and potentially current standard of care. The advent and continued use of proton beam therapy for use in patients with prostate cancer is a slowly evolving topic due to the lack of level 1 evidence demonstrating improved efficacy with its use. Despite this, not all patients are provided with equal access to this treatment modality. While there have been several publications utilizing either single-institutional analyses or national medical databases, to date, a comprehensive review summarizing the disparities and inequities in proton beam therapy use in prostate cancer does not exist. Our research sought to elucidate trends in this topic and create a compendium that can be used to inform future study and equitable advances in the field.

## 1. Introduction

Prostate cancer, the most common cancer in males across the United States, is characterized by well-documented disparities in incidence, management, and outcomes [1,2,3,4]. While management options for prostate cancer continue to expand, radiotherapy (RT) remains an essential curative therapeutic option. Although historical forms of RT have been photon (X-ray) based, proton beam therapy (PBT) is a newer modality touted to be dosimetrically favorable to photon-based therapy [5]. However, the degree to which dosimetric benefits may translate to toxicity reduction remains unclear, which is important considering that PBT delivery costs are greater than photon-based RT [6,7]. While preliminary findings of the PARTIQoL trial showed similar results for the reported bowel summary scores of Expanded Prostate Cancer Index Composite (EPIC) patients between the two modalities at 24 months, grade toxicities and other secondary endpoints have not been reported yet. The COMPPARRE trial is also evaluating these differences, and together, the findings will be deterministic in how we approach prostate cancer [8,9]. Because the implementation of PBT has more recently progressed quickly, there remain several areas requiring elucidation through comprehensive summarization.

Well-documented health disparities and inequities exist in prostate cancer incidence, healthcare resource utilization, and mortality. Providing care that is equitable and that does not vary in quality because of personal characteristics such as gender, ethnicity, geographic location, and socioeconomic status is considered one of the six domains of health care quality according to the National Academy of Medicine [10]. Inequities in the utilization of proton beam therapy (PBT) specifically in prostate cancer have not yet been comprehensively summarized.

Studies have demonstrated that PBT is one of the least commonly used modalities of radiation therapy in men with prostate cancer and that its utilization may vary given challenges in access to care [11,12]. To address this knowledge gap, we performed a review of the published articles on disparities in the delivery of PBT for prostate cancer (Table 1). Our purpose was to review and summarize the reported health disparities and inequities in the use of proton beam therapy for prostate cancer in the United States.

## 2. Materials and Methods

We queried the PubMed search engine for original studies published in English that studied health disparities in the delivery of PBT for prostate cancer. Query terms were as follows: prostate AND proton AND (disparities OR IMRT OR race OR insurance OR socioeconomic OR inequities). Articles known to the authors and those based on the references of the obtained articles were also utilized. Studies were excluded if they did not specifically evaluate PBT, were not based in the United States, or did not directly examine health inequities. Unpublished abstracts were not included owing to the inability to completely assess their validity and methodologies. There were no date restrictions, and the search included articles published through December 2023. It was infeasible to perform a meta-analysis on the available literature because of the inherent heterogeneity of the study topic.

## 3. Results

A total of 162 studies were captured in the initial query. Of these, 22 met the inclusion criteria. These 22 studies, published from 2012 to 2023, comprised 13 population-based analyses, 5 single-institutional analyses, 3 cost/modeling investigations, and 1 survey-based study (Figure 1).

The datasets analyzed in these studies include the National Cancer Database (NCDB), the Surveillance, Epidemiology, and End Results (SEER), and several single-institutional studies. Of the 22 articles, 19 had a primary outcome of PBT utilization, with the other 3 evaluating treatment toxicity, clinical benefit, and participation in clinical trials. The most common health inequities pertained to factors such as age, race/ethnicity, socioeconomic status, insurance type, and geography of the patient and treatment center. Of these, a majority of the studies identified racial disparities impacting the use of PBT for prostate cancer management (Figure 2).

### 3.1. Age

Age was evaluated in 15 of the 22 (68%) studies, and overall showed a decreased likelihood of receiving PBT with advancing age. For example, a study of 143,702 patients in the National Cancer Database (NCDB) sought to examine trends in and characterization of PBT for prostate cancer [13]. The authors found a steady increase in PBT utilization (2.7% of all external-beam RT in 2004 to 5.6% in 2013), which was most pronounced in low-risk cases. The odds ratio (OR) for PBT in patients ≥ 65 years old was 0.74, when accounting for other factors such as the comorbidity index or insurance status in multivariable analysis (*p* < 0.001). These findings are very comparable to the findings of a similar NCDB study of 187,730 patients that also sought to evaluate trends and practice patterns for PBT. That study determined through multivariable analysis that each successive year of age at diagnosis was associated with a 5% lower chance of receiving PBT [14]. From a dataset of Medicare beneficiaries (*n* = 27,647) assessing the same question as the aforementioned NCDB publications, Yu and colleagues found that the percentage of PCa (prostate cancer) patients receiving PBT was 3.3% between the ages of 66 and 69, 2.1% for those 70–74, 1.4% for those 75–79, and 1.0% for those 80–84 [15]. These corresponded to the adjusted ORs, with reference to ages 66–69, of 0.66, 0.45, and 0.33, respectively (*p* < 0.001).

### 3.2. Race and Ethnicity

Regarding race and ethnicity, 13 of the 22 (59%) of the investigations reported that non-White populations were less likely to receive PBT. The aforementioned NCDB studies demonstrated ORs for Black and other non-Black minorities (with reference to White patients) to be 0.66 and 0.65, respectively [13], and 0.20 and 0.57, respectively [14] (*p* < 0.001). These ORs in a single-institution retrospective study (n = 633) aimed at evaluating sociodemographic inequalities in PBT receipt, were 0.29 (*p* < 0.001) and 0.42 (*p* < 0.025), respectively [16]. Parikh-Patel et al. underscored these results when their research showed that those who received PBT therapy were disproportionately White (70.7%). This same cohort consisted of only 5.6% non-Hispanic Blacks (*p* < 0.001) [17]. Unfortunately, these disparities have not waned, as a more recent study utilizing the NCDB found the racial disparity in PBT utilization widening for both ASTRO model policy proton therapy-designated group 1 (PBT use indicated) and group 2 (PBT use still under investigation) cancers, the latter of which includes prostate cancer. The study showed that overall, between 2004 and 2018, Black males were less likely to receive PBT when compared to their White counterparts (OR, 0.67; 95% CI, 0.64–0.71). For group 1 cancers, Black patients were less likely to receive PBT when indicated vs. their White counterparts (0.4% vs. 0.8%; OR, 0.49; 95% CI, 0.44–0.55). For group 2 cancers, racial disparities persisted, as the Black race was a negative predictor of PBT use in comparison to its White counterpart (0.3% vs. 0.4%; OR, 0.75; 95% CI, 0.70–0.80) [18]. To date, there is no evidence to attribute differential rates in adverse events following RT to racial inequities in PBT utilization [19]. A single-institution, matched-pair analysis of health-related quality of life after PBT demonstrated no statistically significant differences in sexual function, urinary incontinence, urinary obstruction, or bowel summary scores between Black and White patients [20]. With respect to clinical trial enrollment, a prospective assessment of patients’ willingness to participate in a hypothetical randomized controlled trial of IMRT versus PBT found that 69% of Black patients endorsed a willingness to enroll, which was numerically higher than White patients (55%) [21]. Of note, the primary motivation for the vast majority (77%) of Black patients for enrollment in such a trial was altruism for other patients, which was numerically different from White patients (45%).

### 3.3. Socioeconomic and Insurance Status

Of the 22 studies, 10 (45%) individually evaluated socioeconomic and insurance status, with most demonstrating that PBT delivery correlates with numerous markers of socioeconomic status. Yabroff et al. showed that PBT use has increased across all insurance types (private vs. uninsured vs. Medicaid vs. Medicare), in group 1 (annual percent changes (APCs) of 20.89 for private insurance, 22.78 for uninsured, 21.01 for Medicaid, and 28.80 for Medicare; *p* < 0.001 for all, 2010–2018) and group 2 (APCs of 32.04 for private insurance, 28.24 for Medicare, 53.01 for Medicaid, and 51.31 for the uninsured; *p* < 0.001 for all, 2014–2018) [22]. The other aforementioned NCDB publications demonstrated that independent predictors of PBT administration included a higher median income, a higher percentage of adults with a high school diploma in the zip code, and a longer travel distance to a PBT center [13,14,16]. While potential confounding factors might stem from the insurance type, as demonstrated in Mendenhall et al.’s study that highlighted the influence of payor type on approval rates (with Medicare notably approving more patients than commercial payors for proton beam therapy for prostate cancer), multivariable analyses were employed to account for insurance-related variables [23,24,25]. These findings were also supported by two single-institutional cost investigations at the MD Anderson Cancer Center and of the Surveillance, Epidemiology, and End Results (SEER) database linked with Medicare [26,33]. Although neither utilized dedicated multivariable analyses, most patients in the PBT cohorts comprised patients with higher median household incomes and/or high school diplomas.

### 3.4. Facility Characteristics

Of the 22 publications, 6 (30%) evaluated this parameter. Although it is intuitive that academic centers are more likely to deliver PBT [14,27], owing to the availability of newer technologies at these facilities, a cost analysis evaluated the novel characteristics of patient volume and reimbursement scenarios. The up-front cost expenditure to build proton centers, compounded by interest rates, creates a large debt for institutions. Reimbursement models, such as the Affordable Care Act, may make it difficult for facilities to receive an adequate return on investment without sacrificing the quality of care (due to the higher volume needed to fund the operational cost of PBT) Based on this study, the authors concluded that fee-for-service centers and those with higher patient volumes would be more associated with financial stability, whereas accountable care reimbursement would necessitate a substantial increase in patient volume in order to remain financially viable. These factors are important to consider, as facility-related disparities impact patients with a wide variety of socioeconomic, racial, and age-based backgrounds [28].

## 4. Discussion

Proton therapy for localized prostate cancer is part of a growing body of technological advances aimed at maximizing tumor control while minimizing side effects on surrounding organs-at-risk, providing an advancing modality for increasingly precise and targeted radiation delivery [34,35,36]. With the increasing utility of PBT throughout the nation and worldwide, it is imperative to better summarize the available evidence of disparities and inequities in receipt thereof. While data from a recent single-center, case-matched, retrospective analysis demonstrated similar outcomes in long-term disease control between PBT and IMRT for prostate cancer, it also indirectly underscored the disparities that exist in PBT utilization, demonstrating that significantly fewer Black or African American patients received PBT for PCa relative to their White counterparts, further necessitating a dire need to elucidate mechanisms of disparities and inequities in our healthcare system [29].

This review of 22 publications demonstrates that substantial age-based, racial, socioeconomic/insurance-related, and facility-associated disparities exist for PBT in prostate cancer. The identification of these disparities provides a framework to better address these, as the utility of PBT across several disease sites continues to expand. Indeed, such disparities have also been noted in the utilization of PBT for other malignancies, including pediatric, where the factors associated with PBT utilization are a younger age, having private/managed care rather than Medicaid or no insurance, a higher median household income and parental educational attainment, or traveling more than 200 miles to the center [30,37].

The findings with regard to age are noteworthy. It is well recognized that age is an independent prognostic factor for most tumors, and the chance of experiencing potential decreases in long-term toxicities (albeit not proven for prostate cancer) inversely correlate with age. Additionally, the lower integral dose afforded by PBT may reduce the secondary malignancy rate, which is of less concern in advancing age but may still be relevant, as demonstrated in a recent NCDB study, which found a lower rate of secondary malignancies in patients with prostate cancer undergoing PBT compared to IMRT [38]. Moreover, it is also well recognized that older patients remain under-represented in cancer clinical trials, although it is uncertain which patients in the studies evaluated herein were treated on protocol [39,40]. Nevertheless, it should be noted that the age differences persisted when adjusting for the fact that older patients receive Medicare (which supports PBT to a greater degree than other insurances). A single-institution retrospective study queried their billing center at a large academic center for patients who were considered for PBT from January 2013 to December 2016 using insurance decision (approval vs. denial) as an endpoint. Using multivariate analysis, Medicare insurance was the strongest predictor of insurance approval with an OR of 14.2 (*p* < 0.001), while secondary insurance approval for treatment had an OR of 2.6 (*p* < 0.001). While this particular study was limited to head and neck cancers, these findings indicate the pervasiveness of this disparity as it extends to other disease sites [31].

The racial findings herein are likely not unifocal but part of a constellation of several factors that reflect the difficulty of obtaining the proper financial clearances (e.g., prior authorization) to receive oncologic therapy (especially for PBT) in the United States. This is best evidenced by evaluating the ORs prior to, and following, multivariable adjustment in several of the analyses herein. These ORs are often quite different, implying that racial disparities are intertwined and mediated by other variables, such as insurance, income, education, and other factors. Mahal et al. study showed that Black, Hispanic, or other minority patients were significantly less likely to receive proton therapy compared to White patients (adjusted odds ratio (AOR), 0.20 (95% confidence interval (95% CI), 0.18–0.22; *p* < 0.0001), AOR, 0.57 (95% CI, 0.48- 0.66; *p* < 0.0001), and AOR, 0.59 (95% CI, 0.50–0.69; *p* < 0.0001), respectively) [14]. Moreover, a recently published cross-sectional study highlighted that age (>65 years old) and socioeconomic status were negative determinants for access to PBT, further underscoring the multifactorial nature of these findings [41]. Notably, not all of these issues can be adequately addressed by multivariable analyses in retrospective studies, and therefore, it still appears that race has a strong independent impact on PBT utilization, potentially related to intrinsic biases and concerns. Further work to address these notions is needed.

Socioeconomic factors are also important to evaluate but suffer from a lack of adequate granularity in most of the available data. For instance, Medicaid reimbursement varies by state/region [42], and not all private insurers are equally willing to reimburse for a given RT technique [43]. To this extent, the analysis by Elnahal et al. is important, in that they modeled reimbursement scenarios representative of the current economic landscape (i.e., fee-for-service versus accountable care) [27]. As shown in their study, the affordability of PBT for not only proton centers/institutions but also patients of all backgrounds could be significantly impacted by ongoing policy changes and disparately impact certain demographic groups over others. Furthermore, this disparity may be exacerbated by denials and the requirement for prior authorization by insurance companies, which, as Gupta et al. showed, lead to significant delays in patient care (average of ~3 weeks). The study underscored the necessity for a streamlined process and agreement between payors and providers in which coverage policies conform to progressing national guidelines [32].

The limitations of this analysis include mainly the search term methodology, which may have excluded older studies due to the evolution of language and terminology in medicine. The scholarly works referenced in this manuscript have their own inherent specific limitations that include the sample size, length of study, and specific parameters utilized by the researchers.

## 5. Conclusions

Although PBT utilization is rapidly expanding, not all patients may experience this to an equal degree. This review demonstrates that substantial age-based, racial, socioeconomic/insurance-related, and facility-associated disparities exist for PBT in prostate cancer. The identification of these disparities provides a framework to better address these inequities as the utility of PBT across several disease sites continues to expand. As the utilization of proton therapy increases across various and new disease sites and expands across the country and globally, it is important to assess disparities and inequities in its utilization in order to address and improve equitable access so that all patients may benefit from the study and delivery of protons.

## Figures and Tables

**Figure 1 cancers-16-03837-f001:**
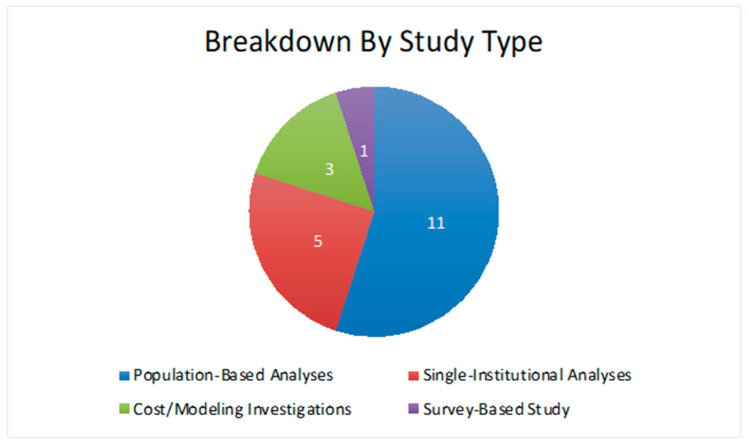
Breakdown of study characteristics.

**Figure 2 cancers-16-03837-f002:**
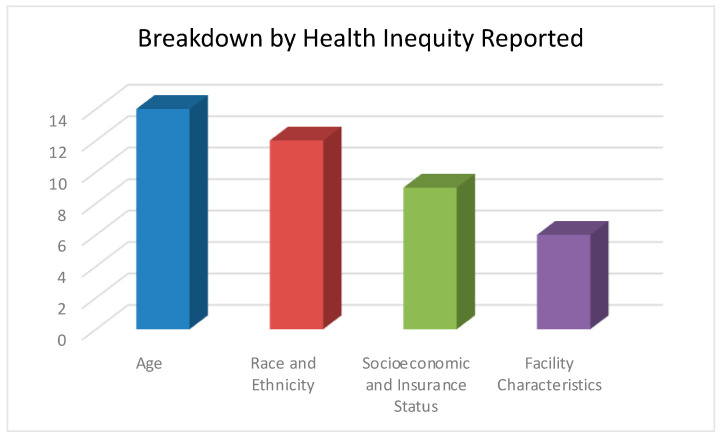
Breakdown of health inequities reported.

**Table 1 cancers-16-03837-t001:** List of studies meeting the inclusion criteria regarding health disparities and inequities in the utilization of proton therapy for prostate cancer.

Reference #	Author and Year	Study Title	Study Type, Cohort, (Practice Setting)	Sample Size	Population	Key Finding(s)
[11]	Agrawal et al. (2022)	Pattern of Radiotherapy Treatment in Low-Risk, Intermediate-Risk, and High-Risk Prostate Cancer Patients: Analysis of National Cancer Database.	Retrospective; population-based analyses (National Cancer Database)	199,926	Men with PCa diagnosed between 2004 and 2015, PSA of 0.2–97.9 ng/mL, GS of 2–10, clinical stage defined as 1, 2, 3, 4, 2A, or 2B. AJCC N0 and M0	This study revealed that IMRT was the most common treatment modality for PCa patients. Brachytherapy, SBRT, and IMRT+BT exhibited similar survival rates, whereas proton showed slightly better overall survival across the three risk groups.
[12]	Mukherjee et al. (2021)	Trends and variations in utilization and costs of radiotherapy for prostate cancer: A SEER Medicare analysis from 2007 through 2016.	Retrospective; population-based analyses (SEER *)	51,686	Men diagnosed with PCa between 2007 and 2015	For Medicare beneficiaries with a first-time diagnosis of prostate cancer, the utilization of (IMRT), proton therapy, and SBRT increased over this time period. Brachytherapy decreased. The cost per beneficiary decreased. Age, registry region, and Gleason score were all associated with expenditures.
[13]	Amini et al. (2017)	Patient characterization and usage trends of proton beam therapy for localized prostate cancer in the United States: A study of the National Cancer Database.	Retrospective; population-based analyses (National Cancer Database)	5709	Men with localized (N0, M0) prostate cancer diagnosed between 2004 and 2013, treated with EBRT, with available data on EBRT modality (photon vs. PBT).	PBT for men with localized prostate cancer significantly increased in the United States from 2004 to 2013. Significant demographic and prognostic differences between those men treated with photons and protons were identified.
[14]	Mahal et al. (2016)	National Trends and Determinants of Proton Therapy Use for Prostate Cancer: A National Cancer Database study	Retrospective; population-based analyses (National Cancer Database)	187,730	Men diagnosed with nonmetastatic prostate cancer from 2004 through 2012 who received external beam radiotherapy as their initial form of definitive therapy.	Proton therapy for PCa was much more likely to be delivered at an academic compared to a nonacademic center and to patients who were White, younger, healthier, from metropolitan areas, from zip codes with higher median household incomes, and those who did not have an advanced stage of or high-grade disease. Black and Hispanic males were less likely to receive PBT than White male.
[15]	Yu et al. (2013)	Proton Versus Intensity-Modulated Radiotherapy for Prostate Cancer: Patterns of Care and Early Toxicity	Retrospective; population-based analyses (Chronic Conditions Warehouse)	553	Medicare beneficiaries aged equal to or greater than 66 years who received PRT or IMRT for prostate cancer during 2008 and/or 2009.	Patients receiving PBT were younger, healthier, and from more affluent areas than patients receiving IMRT. There was no statistically significant difference in gastrointestinal or other toxicity at 6 months or 12 months post-treatment.
[16]	Woodhouse et al. (2017)	Sociodemographic disparities in the utilization of proton therapy for prostate cancer at an urban academic center	Retrospective; single academic institution (academic)	663	All patients with low- and intermediate-risk prostate cancer who underwent definitive radiation therapy between 2010 and 2015.	Patients who underwent IMRT were more likely to be older, Black, and living in poverty or close to the facility. Patients of Black and other races were less likely to receive PT compared to patients of White race.
[17]	Parikh-Patel et al. (2020)	A population-based assessment of proton beam therapy utilization in California	Retrospective; population-based analyses (California Cancer Registry)	8609	Persons with diagnoses of all cancer types from 2003 to 2016, inclusive, who had any type of RT were identified in the California Cancer Registry in this retrospective analysis.	Patients with cancer with Medicare insurance coverage were more likely to receive proton beam therapy compared to patients with private insurance. Compared to non-Hispanic Whites, all other racial/ethnic groups had significantly lower odds of being treated with proton beam therapy, across various cancer types, after accounting for other relevant demographic and clinical factors.
[18]	Nogueira et al. (2022)	Association of Race With Receipt of Proton Beam Therapy for Patients With Newly Diagnosed Cancer in the US, 2004–2018	Retrospective; population-based analyses (National Cancer Database)	5,225,929	Black and White individuals diagnosed with PBT-eligible cancers between 1 January 2004 and 31 December 2018 in the National Cancer Database.	Black patients were less likely to be treated with PBT than their White counterparts. Racial disparities were greater for group 1 cancers than group 2 cancers. Racial disparities in PBT receipt among group 1 cancers increased over time.
[19]	Sheets et al. (2012)	Intensity-modulated radiation therapy, proton therapy, or conformal radiation therapy and morbidity and disease control in localized prostate cancer	Retrospective; population-based analyses (SEER *)	12,976	Men diagnosed with non-metastatic PCa from 2000 to 2009.	In patients with nonmetastatic prostate cancer, the use of IMRT, compared to conformal radiation therapy, was associated with less gastrointestinal morbidity and fewer hip fractures but more erectile dysfunction; compared to proton therapy, IMRT was associated with less gastrointestinal morbidity.
[20]	Bryant et al. (2016)	Does Race Influence Health-Related Quality of Life and Toxicity Following Proton Therapy for Prostate Cancer?	Retrospective; single academic institution (academic)	1536	Men diagnosed with clinically localized PCa and treated from 2006 to 2009 with definitive proton therapy to a median dose of 78 Gy +/androgen deprivation therapy.	No difference was found in the Expanded Prostate Index Composite 26-question sexual summary or the Urinary Incontinence Index Composite 26-question sexual summary between the 2 groups, nor was there a difference in grade 2 or higher GI toxicity. AAs had a statistically nonsignificant higher absolute incidence of late grade 3 genitourinary toxicity.
[21]	Shah et al. (2012)	Prospective Preference Assessment of Patients’ Willingness to Participate in a Randomized Controlled Trial of Intensity-Modulated Radiotherapy Versus Proton Therapy for Localized Prostate Cancer	Prospective cohort study; single institution (academic)	46	Men with clinically localized PCa and aged >18 years with histologically confirmed PCa and clinical T1c-T2b stage disease between October 2010 and April 2011.	Twenty-one factors impacted patients’ willingness to participate (WTP), which largely centered on five major themes: altruism/desire to compare treatments, randomization, deference to physician opinion, financial incentives, and time demands/scheduling. A substantial proportion of patients indicated high WTP in a RCT comparing IMRT and PBT for PCa.
[22]	Nogueira et al. (2022)	Assessment of Proton Beam Therapy Use Among Patients With Newly Diagnosed Cancer in the US, 2004–2018	Retrospective; population-based analyses (National Cancer Database)	5,919,368	Individuals newly diagnosed with cancer between 2004 and 2018 were selected from the National Cancer Database.	PBT use increased in the US between 2004 and 2018; prostate was the only cancer site for which PBT use decreased temporarily between 2011 and 2014, increasing again between 2014 and 2018.
[23]	Mendenhall et al. (2021)	Insurance Approval for Definitive Proton Therapy for Prostate Cancer	Retrospective; single institution (academic)	1592	Patients with localized prostate cancer.	On multivariate analysis, factors affecting PT approval for prostate treatment included coverage of PT per policy, insurance type, and time: Proton insurance approval for prostate cancer had decreased, was most influenced by the type of insurance a patient belonged to, and was unrelated to clinical factors (risk group) in this study.
[24]	Pan et al. (2017)	Adoption of Radiation Technology Among Privately Insured Nonelderly Patients With Cancer in the United States, 2008 to 2014: A Claims-Based Analysis	Retrospective; claims-based analyses (MarketScan Commercial Claims and Encounters database)	8,040,459	Radiotherapy utilizationbetween 2008 and 2014 (IMRT, proton, brachytherapy, or stereotactic).	Conventional RT and IMRT were the most commonly used technologies by far. Proton radiation was used most commonly for prostate cancer, although it decreased over a time period.
[25]	Sharma et al. (2019)	Patient Prioritization for Proton Beam Therapy in a Cost-Neutral Payer Environment: Use of the Clinical Benefit Score for Resource Allocation	Prospective; single institution (academic)	205	Patients considered for PBT at an academic institution who were prospectively scored using the CBS.	Multivariate analysis adjusting for insurance status revealed both the clinical benefit score (CBS) and insurance to be significant predictors of receiving PBT. CBS utilized was significantly associated with the receipt of PBT in a cost-neutral payer setting.
[26]	Halpern et al. (2016)	Use, Complications, and Costs of Stereotactic Body Radiotherapy for Localized Prostate Cancer	Retrospective; population-based analyses (SEER *)	34,397	Men who underwent SBRT, intensity-modulated radiation therapy (IMRT), brachytherapy, and proton beam therapy as primary treatment for prostate cancer during 2004 and 2011 from Surveillance, Epidemiology, and End-Results Program (SEER)-Medicare linked data.	The utilization of SBRT and proton therapy for localized prostate cancer has increased over time. Despite men of a lower stage undergoing SBRT, it was associated with greater toxicity but lower healthcare costs compared to IMRT and proton therapy.
[27]	Tang et al. (2021)	Influence of Geography on Prostate Cancer Treatment (2021)	Retrospective; population-based analyses (National Medicare Database)	89,902	Men diagnosed and treated for prostate cancer in 2011–2014.	Established providers using IMRT, prostatectomy, and brachytherapy were predominantly based in major urban centers. Rural areas had reduced numbers of providers utilizing brachytherapy. Greater distance was associated with a decreased probability of treatment.
[28]	Elnahal et al. (2013)	Proton Beam Therapy and Accountable Care: The Challenges Ahead	Cross-sectional design; operational reimbursement data from a single academic institution	-	The total number of patients able to be seen in a given day was calculated based on the relative time required for complex cases versus simple, prostate, and short prostate cases.	Debt-financed PBT centers face steep challenges to remain financially viable after ACO implementation. Paradoxically, reduced reimbursement for noncomplex cases require PBT centers to treat more such cases over cases for which PBT has demonstrated superior outcomes. Relative losses are highest for those facilities focused primarily on treatment.
[29]	Bao et al. (2023)	Case-Matched Outcomes of Proton Beam and Intensity-Modulated Radiation Therapy for Localized Prostate Cancer	Prospective cohort study; single institution (academic)	334	Patients with clinically localized PCa who underwent definitive PBT or IMRT at the Hospital of University of Pennsylvania between 2010 and 2012 were enrolled on institutional review board–approved prospective protocols evaluating outcomes of conventionally fractionated (CF) (the standard at the time) PBT or IMRT.	Both PBT and IMRT offer excellent long-term disease control for PCa with no significant differences between the 2 modalities in BFFS, PCSS, and OS in matched patients. In the unmatched cohort, fewer incidences of secondary malignancy were noted in the PBT group; however, owing to an overall low incidence of secondary cancer and imbalanced patient characteristics between the 2 groups, these data are strictly hypothesis-generating and require further investigation.
[30]	Waddle et al. (2017)	Photon and Proton Radiation Therapy Utilization in a Population of More Than 100 Million Commercially Insured Patients	Retrospective; population-based analyses (OptumLabs Data Warehouse)	474,533	Privately insured and Medicare Advantage enrollees in the United States. PBRT users andtheir variation over an 11-year period.	This study is the largest and most geographically diverse description of RT utilization to date. Proton beam utilization remains low and has had little impact on overall utilization compared to IMRT. The utilization rate for pediatric patients remains low, and the greatest change in RT use was the increase in IMRT for prostate cancer.
[31]	Ning et al. (2019)	The Insurance Approval Process for Proton Radiation Therapy: A Significant Barrier to Patient Care	Cross-sectional design; single academic institution billing database	1753	Patients with thoracic or head and neck (HN) cancer considered for proton therapy from January 2013 through December 2016.	The insurance process is a resource-intensive barrier to patient access associated with significant time delays. Medicare was the strongest predictor of initial insurance approval.
[32]	Gupta et al. (2019)	Insurance Approval for Proton Beam Therapy and its Impact on Delays in Treatment	Retrospective; single institution (academic)	444	Patients considered for PBT between 2015 and 2018 at a National Cancer Institute designated Comprehensive Cancer Center.	Prior authorization requirements in adults represent a significant burden in initiating PBT and cause significant delays in patient care. Insurance approval is arbitrary and has become more restrictive over time, discordant with national clinical practice guidelines. Payors and providers should seek to streamline coverage policies in alignment with established guidelines to ensure appropriate and timely patient care.

* SEER—Surveillance, Epidemiology, and End Results.

## Data Availability

Our data can be found on PubMed by querying prostate AND proton AND (disparities OR IMRT OR race OR insurance OR socioeconomic OR inequities).

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
