# Peer review of "Health Disparities and Inequities in the Utilization of Proton Therapy for Prostate Cancer"

_cancers, 2024, doi:10.3390/cancers16223837_

Round 1
Reviewer 1 Report
Comments and Suggestions for Authors
The authors conduct a non-systematic review of the scientific literature on health disparities and inequities in the utilization of proton therapy for prostate cancer in the United States. The work adds nothing to what is already known and that is that in the U.S. the poor, blacks, Hispanics and the old and children cannot get treatment if they are poor. The authors should consider it a shame that “Although PBT utilization is rapidly expanding, not all patients may experience this to an equal degree.”
It is unclear how such a study, although internally consistent, would allow for an overcoming of these inequities even though the authors state, “Identification of these disparities provides a framework to better address these as the utility of PBT continues to expand 34 across the US and globally.”
For these reasons, it is believed that only a universalistic health care system can solve the problem posed by the authors and that the study conducted is of no use.
Author Response
Thank you for highlighting this. Knowing a problem exist does not disqualify echoing it's prognostic potential and ramifications. In fact, we believe the purpose of this review is to serve as stepping stone for the next phase in addressing the root causes of the disparities that exist in prostate cancer care. A key component in root cause analysis is gathering data, and we believe our compendium of the expansive data makes tackling this issue achievable.
Reviewer 2 Report
Comments and Suggestions for Authors
This is a review on papers related to disparities related to proton beam therapy in the treatment of prostate cancer. The authors conducted a PubMed search through 12/2023 looking for original publications examining disparate utilization of proton beam therapy (PBT) for prostate cancer.
They found 22 studies meeting their inclusion criteria including 13 population-based analyses and 5 single-institutional analyses. The most common health inequities pertained to factors such as age, race/ethnicity, socioeconomic status, insurance type, and geographical location of the patient and treatment center.
Overall, they found that older patients were less likely to receive PBT. Several studies also found that non-white patients had a lower likelihood of receiving PBT than did whites. Conversely, patients with a higher median income and higher education level had an increased likelihood of receiving PBT.
This is a very straightforward review of existing articles that assess disparities and their relationship to the likelihood of receiving proton beam therapy. It is s descriptive compendium of existing studies. The conclusions are sound.
The only suggestion I have is to include a reference to an abstract presented at the recent ASTRO national meeting at which an important study was presented (PARTQoL) that showed equivalent local control between patients treated with proton or photon beam therapy but failed to demonstrate any improvement in bowel toxicity. Although this study did not address disparities, it seems highly relevant to this review of studies using proton beam therapy for prostate cancer.
Author Response
Comments 1: The only suggestion I have is to include a reference to an abstract presented at the recent ASTRO national meeting at which an important study was presented (PARTQoL) that showed equivalent local control between patients treated with proton or photon beam therapy but failed to demonstrate any improvement in bowel toxicity. Although this study did not address disparities, it seems highly relevant to this review of studies using proton beam therapy for prostate cancer.
Response 1: Thank you for your comment. We agree, this information should be included in our review as supporting evidence as it is preliminary evidence. The edit can be found Page 2, paragraph 1, lines 46-50.
Reviewer 3 Report
Comments and Suggestions for Authors
The current manuscript, "Health Disparities and Inequities in the Utilization of Proton Therapy for Prostate Cancer," has been reviewed and found to be well-structured and insightful. I recommend this for publication.
- The study presents a comprehensive analysis of health disparities and inequities in proton beam therapy (PBT) for prostate cancer, incorporating data from 22 relevant studies (from 162 studies searched from pubmed). Authors defined methodology clearly, using specific query terms to extract relevant studies, ensuring a thorough review of the literature.
- Use of diverse datasets such as the NCDB and SEER strengthens the findings, providing credible and well-supported discussion/conclusions.
- The identification of key disparities, including age, race, socioeconomic status, insurance type, and facility characteristics, provides valuable insights into barriers to proton therapy access.
- Socioeconomic factors like income and education are well-highlighted, showing their significant role in determining PBT access.
- Manuscript successfully ties the clinical relevance of proton therapy dosimetric advantages to the need for equitable access to this treatment modality
Author Response
We appreciate your comments and recommendation.
Reviewer 4 Report
Comments and Suggestions for Authors
Dear authors,
your paper addresses an interesting topic, which could become prominent in the coming years, probably pervaded by an increasing availability of proton therapy worldwide.
I have no particular issues to raise. You could contextualize your research field even better by adding the following sentence at the beginning of the discussion:
"Proton therapy for localized prostate cancer is part of a growing body of technological advances aimed at maximizing tumor control while minimizing side effects on surrounding organs-at-risk, and is emerging as one of the most exciting solutions for increasingly precise and targeted radiation delivery [a, b, c]".
and cite PMID: 33813420, PMID: 32248826 and PMID: 30324346 in [a, b, c].
In my opinion, these sentence and references provide a solid background to your topic.
Author Response
Comment 1: "Proton therapy for localized prostate cancer is part of a growing body of technological advances aimed at maximizing tumor control while minimizing side effects on surrounding organs-at-risk, and is emerging as one of the most exciting solutions for increasingly precise and targeted radiation delivery [a, b, c]". and cite PMID: 33813420, PMID: 32248826 and PMID: 30324346 in [a, b, c].
Response 1: Thank you for your comment. We agree with this comment. Therefore, we have added this change to Page 11, paragraph 1, lines 189-192
Round 2
Reviewer 1 Report
Comments and Suggestions for Authors
The authors have not mentioned my criticism neither to contest them.
Author Response
Comments 1: It is unclear how such a study, although internally consistent, would allow for an overcoming of these inequities even though the authors state, “Identification of these disparities provides a framework to better address these as the utility of PBT continues to expand 34 across the US and globally.”
Response 1: Thank you for pointing this out. We agree with this comment. Therefore we have expanded on this idea on page 13, paragraph 4, lines 290-294.